# Changes in Undergraduate Students’ Psychological Well-Being as They Experience University Life

**DOI:** 10.3390/ijerph16162864

**Published:** 2019-08-10

**Authors:** Xinqiao Liu, Siqing Ping, Wenjuan Gao

**Affiliations:** 1Graduate School of Education, Peking University, Beijing 100871, China; 2China Institute for Educational Finance Research, Peking University, Beijing 100871, China; 3Department of Public Policy & Management, Guangzhou Administration School, Guangzhou 510070, China

**Keywords:** undergraduate students, well-being, longitudinal study, depression, anxiety, stress

## Abstract

The onset of most lifetime mental disorders occurs during adolescence, and the years in college, as the final stage of adolescence in a broad sense, deserve attention in this respect. The psychological well-being of undergraduate students can influence not only their academic and professional success, but also the development of society as a whole. Although previous studies suggested psychiatric disorders are common in the adult population, there was little consistent information available about undergraduate students’ mental health problems. This research aimed to describe the changes in depression, anxiety, and stress of Chinese full-time undergraduate students as they experienced university life using the Depression Anxiety Stress Scales-21 (DASS-21). The main conclusions of our study were as follows: (1) on average, students’ severity scores of depression during the four academic years varied between 7.22 and 7.79, while stress scores ranged from 9.53 to 11.68. However, the anxiety scores of college students in the first three years turned out to be 7.40, 7.24 and 7.10, respectively, slightly overtaking the normal threshold of 7. These results indicated that Chinese college students, in general, were mentally healthy with regard to depression and stress, but their average anxiety levels were beyond normal in the first three years. (2) As for the proportions of students with different degrees of severity, approximately 38% to 43% of college students were above the normal level of anxiety, about 35% above the normal level of depression, and around 20% to 30% above the normal level of stress. (3) There were significant differences in the psychological health states of students of different years, especially among the sophomores, juniors, and seniors; the highest score of depression, anxiety, and stress all appeared in the first or second year on average, but some improvements were achieved in the third and last years. The findings suggested that colleges and universities need to pay special attention to psychologically unhealthy students, and with concerted efforts by the government, formulate mental health policies in the prevention, detection, and treatment of students’ psychiatric disorders, rather than just focusing on their average levels of mental health.

## 1. Introduction

Mental health serves as an integral component of overall health and well-being, and plays an important role in maintaining physical health. Previous studies also found that adverse physical health status (e.g., obesity, asthma, skin diseases, hepatitis, heart diseases) had an impact on mental health in young people and adults [1,2,3,4,5]. As part of mental health, emotional well-being incorporates happiness, interests in life, satisfaction and quality of life [6,7,8], and the ability to recognize, express and regulate one’s emotions [9]. Negative emotions often manifest as depression, anxiety, irritability, excitement, etc. The widely recognized Depression Anxiety Stress Scales-21 (DASS-21) is an effective measure of negative emotions in Chinese people [10,11]. Depression is typically characterized by melancholy, frustration, and anguish, while anxiety refers to unnecessary tension for objective things and interpersonal relationships; general distress is the common feature for both depression and anxiety [12]. Indeed, depression, anxiety, and stress are deemed as significant indicators of psychological well-being [13].

In the past decades, the number of people suffering from depression and anxiety have increased sharply, which has posed a grave threat to society. Relevant research found that the global aggregate point prevalence of depression was 12.9 percent, by combining the data of over 1 million participants from 30 countries between 1994 and 2014. South America had the highest depression rate, reaching up to 20.6 percent, followed by Asia (16.7%), North America (13.4%), Europe (11.9%), Africa (11.5%), and Australia (7.3%) [14]. Meanwhile, depression and anxiety problems are more prevalent among college students [15,16,17]. A meta-analysis study indicated that the worldwide prevalence of depression among medical students was 34.0 percent, with the highest proportion reported in Asia at 43.0 percent [18]. Another related study showed that in Asia, 11.0 percent of college students suffered from depression, while the aggregate prevalence of anxiety disorders was 7.04 percent [19]. In China, college students have become psychologically disadvantaged, with roughly one-fifth of students struggling with different levels of mental problems, and a considerable number of students experiencing depression, anxiety, and stress [20].

College is a crucial period of life for students to shape proper values, worldviews, outlooks of life, and resilience [21]. This period, in some cases, is defined as the final stage of adolescence. Adolescence is the phase of life stretching between childhood and adulthood, and rather than referring to the 10–19 age group, a definition of adolescence as 10–24 years old corresponds more closely to popular understandings of this life stage [22]. College students’ mindset not only determines their academic achievement in school, but also predicts their adaptability to the workplace and society in the future. Therefore, it is of great significance to clarify the influencing factors and mechanism of college students’ psychological well-being in various backgrounds. College students’ emotional well-being was significantly correlated with their monthly household income [13], ethnicity [13], social life [13,23,24], parental education and occupations [23], interests in major [25], hometown [23], body image [23,25], female sex [23], age [23], socioeconomic circumstance [23], academic performance [26], the pressure to succeed [26], post-graduation plans [25], and financial difficulties [27,28,29]. Additionally, some studies found that college students were confronted with multiple kinds of pressure from role changes, study tasks, interpersonal relationships, employment, etc. [30,31]. If this pressure is not relieved in time, this emotional suppression can easily lead to mental disorders. In turn, these negative emotions may affect students’ physical health, academic performance, learning efficiency, as well as lifestyle, or even provoke social isolation and misbehavior [32,33,34,35]. Worse still, mental illness stigma hinders students from seeking psychiatric help, which undoubtedly exacerbates their psychological well-being problems [15,36,37].

Most of the previous studies used cross-sectional data, while some research employed longitudinal designs to study college students’ psychological well-being status, as well as its determinants, in different years. Globally, Puthran et al. (2016) found that medical freshman students had the highest rates of depression at 33.5 percent, which then experienced a significant decline over time to 20.5 percent before graduation [38]. A similar changing trend was witnessed in the context of the United Kingdom. Bewick et al. (2010) pointed out that compared to the pre-university stage, students struggled with the highest levels of strain in the first semester of year one; and there was a significant reduction in levels of distress from semester one to semester two in both the first and third years [39]. Andrews & Wilding (2004) found that 36 percent of previously depressed or anxious students had recovered [27]. Nevertheless, studies in the United States reached the opposite conclusion; that the psychological well-being of students seemed to worsen over time. Beiter et al. (2015) found that compared with freshmen, juniors and seniors scored higher on depression, anxiety and stress scales [26]. Likewise, Rosal et al. (1997) concluded that students’ psychological status resembled that of the general population upon enrolment, but their depression scores experienced a persistent rise over time [40].

In general, existing literature regarding the psychological distress of college students mainly used cross-sectional data, and the few longitudinal studies often had a small sample size, with participants mostly from a single university. Thus, there is a dearth of detailed analysis and comparisons of the mental health status of college students over the four-year span, especially in a Chinese context. Indeed, college students of different years may be exposed to different learning circumstances and experience various degrees of depression, anxiety, and stress. It is critical for colleges to understand students’ emotional changes in order to offer proper guidance and support. On this base, our study followed a cohort of undergraduates from 15 Chinese universities; using descriptive statistics and multiple group analysis, we pictured the changing trends of students’ negative emotions during their four academic years in college.

## 2. Methods

### 2.1. Participants and Procedure

This study employed data from the “Beijing College Student Panel Survey” (BCSPS) of the “China Education Panel Survey” (CEPS). The sampling frame of this survey was the students’ status data bank provided by the Beijing Municipal Commission of Education. Participants of the survey were randomly chosen from students admitted to 15 universities in the years of 2006 and 2008, and these two cohorts were tracked for four consecutive years from June 2009 to June 2012. The first three rounds were carried out on-site with the cooperation of Beijing Municipal Commission of Education and the Office of Student Affairs of the universities; thereafter, in year four, the participants were invited via text messages and e-mails to log in to the questionnaire website with a unique code, and this round of investigation was conducted entirely online [41]. In order to explore changes in students’ mental health states across years during college, the 2008 cohort was selected and analyzed in our study. The initial numbers of participants from different universities in the first round have been listed in Table 1, and the follow-up rate of the latter three rounds was 95.27%, 94.66%, and 90.58%, respectively, with very few sample losses [41]. After removing invalid questionnaires, the effective sample size of this panel study was 1401, including 650 females and 751 males. The proportion of samples from different universities are also shown in Table 1. As for the academic disciplines of students, we have mainly divided their majors into four categories, namely social sciences, humanities, science and engineering and the others (including students who had no specific major upon entering college and a small number of unanswered participants). There were 426 students and 193 students majoring in social science and humanities, respectively accounting for 30.41 percent and 13.78 percent of the sample. Moreover, 757 students studied the discipline of science and engineering, representing 54.03 percent of the total; and another 25 students were classified as others (1.78%).

### 2.2. Measure

In our study, students’ psychological well-being was assessed using the Depression Anxiety Stress Scales-21 (DASS-21), a self-report measure of the severity of three related negative emotional states. The DASS-21 has been widely recognized for its reliability and different forms of validity in a range of studies from different countries with different samples [42,43]. Each of the three DASS-21 scales is measured with seven items. Specifically, “the depression scale assesses the state of dysphoria, hopelessness, devaluation of life, self-deprecation, lack of interest/involvement, anhedonia and inertia; the anxiety scale assesses autonomic arousal, skeletal muscle effects, situational anxiety, and subjective experience of anxious affect; the stress scale is sensitive to levels of chronic non-specific arousal, and it assesses difficulty relaxing, nervous arousal, and being easily upset/agitated, irritable/over-reactive and impatient” [44]. Scores for depression, anxiety, and stress are calculated by summing the scores of corresponding items, which represent the degrees of depression, anxiety, and stress of the subject. It should also be noted that DASS-21 is a short version derived from the basic DASS questionnaire; thus, in order to compare with the cutoff values of conventional severity ratings (see Table 2) [45], the scores on DASS-21 need to be multiplied by two. For this study, the scale reliability coefficients of depression, anxiety, and stress were 0.813, 0.766, and 0.812, respectively, indicating good validity of the DASS measurement.

## 3. Results

### 3.1. Average Scores in Depression, Anxiety, and Stress

Table 3 lists the means and standard deviations of each item in the DASS-21 questionnaire, and in order to interpret the scores, the summed means of the three DASS scales have been multiplied by two, as mentioned above. On average, students’ severity scores of depression during the four academic years varied between 7.22 and 7.79, while the stress scores ranged from 9.53 to 11.68. After comparing with the cutoff values in Table 2, it was evident that college students, in general, were mentally healthy with regard to depression and stress scales. Nevertheless, the anxiety scores in the first three years turned out to be 7.40, 7.24 and 7.10, respectively, slightly overtaking the normal threshold of 7. In the senior year, anxiety seemed to be relieved to some degree, with the mean score falling to 6.63. On average, Chinese college students suffered from above-normal levels of anxiety in the first three years, but they stayed mentally healthy with mean scores of depression and stress in the normal range.

Regarding the differences in the mental states of college students across years, Figure 1 compared their original average scores in depression, anxiety, and stress. For the anxiety scale, freshman students scored 3.70 on average, and the scores decreased gradually from year one to year four, indicating that this situation improved over time. Sophomore students endured relatively high levels of depression and stress, with mean scores reaching up to 5.84 and 3.89, respectively. It was noteworthy that throughout college, students suffered more from stress compared with depression and anxiety. In sum, the mental states of college students deteriorated from the freshman to sophomore year, with some improvements in the last two years.

### 3.2. Proportion of Students’ Depression, Anxiety, and Stress

Figure 2 presents the proportion of participants whose answers on the DASS-21 indicated a normal, mild, moderate, severe, or extremely severe amount of depression over time. We noticed that more than 60 percent of students had a normal level of depression in the four years of college, while 12 to 17 percent of students suffered from moderate depression. A larger proportion of students reported having mild depression in their sophomore year while moderate depression was more likely to be seen in the senior year. About two to five percent of students experienced severe depression, and less than two percent of students struggled with extremely severe depression. It was evident that sophomores had a higher risk of suffering from depression.

Figure 3 shows participants’ responses to their anxiety levels. Students with normal levels of anxiety represented more than 55 percent of the total. However, 8 to 13 percent of students suffered from mild anxiety, and around 20 percent had moderate anxiety. Students with severe or extremely severe anxiety constituted four to six percent. The percentage of students experiencing anxiety was noticeably high in their freshmen year while a larger ratio of junior students had extremely severe anxiety. In Figure 4 which reveals college students’ levels of stress, over 70 percent of students in college experienced normal levels of stress. Students who suffered from moderate stress accounted for approximately 7 percent to 12 percent, while two to five percent struggled with severe stress on campus, and less than 1.1 percent had extremely severe stress. It was apparent that students in their sophomore year suffered the most from stress compared to other years.

With the above analysis, we can conclude that Chinese college students were mentally healthy on average, while the psychological health states of freshmen and sophomores warrants more attention. Descriptive statistics showed that students in their first year of college tended to suffer more from anxiety, while in the sophomore year they experienced relatively high levels of stress and depression. As they entered the last two years of college life, their situation seems to have taken a favorable turn. As for the proportions of students with different degrees of mental health problems, approximately 38 to 43 percent of college students had above the normal levels of anxiety, while around 35 percent had above the normal levels of depression. In addition, students whose stress scores surpassed the clinical cutoff values constituted about 20 to 30 percent of the total. The onset of most lifetime mental disorders occurs before young adulthood [46], and the main groups at this age are college students in many countries. China’s higher education has entered the stage of popularization. According to the overview of educational achievements in China in 2017, the gross enrollment rate of higher education rose to 45.7 percent [47]. Our study indicated that colleges and universities need to pay special attention to psychologically unhealthy students, rather than just focusing on their average levels of mental health.

### 3.3. Multiple Group Analysis

Considering the different academic and employment pressures confronted by students of different years in college, there still needs to be a greater exploration than merely descriptive statistics to conclude whether there exist significant differences in the psychological well-being of college students across years. Multiple group analysis is used to determine whether parameters of a measurement model and/or a structural model are invariant across two or more groups [48]. The invariance testing by the measurement model shows whether the items mean the same thing to the respondents of different groups, while the structural model indicates whether the structural paths are the same across groups [49]. We followed a four-step procedure to determine whether the invariant factorial structure would hold over time. First, an unconstrained model was specified freely estimating all parameters across the four academic years. After this baseline model, equality constraints were applied increasingly for testing. Specifically, at the second stage, the invariance test started by constraining all factor loadings to be equal; while in the next step, we further analyzed with the structural covariances model, holding both the factor loadings and variances equal. Finally, the measurement residuals model constrained the estimates for all factor loadings, variances, and residuals to be equal across years. In this way, this model can be considered as the strictest model for group invariance in our analysis.

Table 4 presents the standardized parameter estimates of the four models across years in college. The multi-sample analysis with the unconstrained model was identified. According to the base model, the λ1, λ2, and λ3 increased gradually from year one to year four, while the general trends of e1, e2, and e3 were downward, revealing depression, anxiety, stress to be better measures of psychological well-being for students over time. The second model, holding all factor loading invariants, proved to be acceptable (χ^2^ (6) = 43.823, *p* < 0.05; NFI = 0.995; NNFI = 0.991; CFI = 0.995; RMSEA = 0.034). The χ^2^ difference test between the baseline model and the measurement weights model was significant (χ^2^ (6) = 43.823, *p* < 0.05), suggesting that the factor loadings of all year groups were different. In the subsequent model, both factor loadings and variances were constrained equally across years, and the result showed that this was also acceptable (χ^2^ (9) = 91.346, *p* < 0.05; NFI = 0.989; NNFI = 0.986; CFI = 0.99; RMSEA = 0.04). The χ^2^ difference test between these two constrained models was also significant (χ^2^ (3) = 47.522, *p* < 0.05). This suggested that, apart from the factor loadings, unique variances of each item were also different in different academic years. Finally, by holding factor loadings, variances and residuals all equal, the multi-sample analysis revealed this constrained model was acceptable (χ^2^ (18) = 261.342, *p* < 0.05; NFI = 0.968; NNFI = 0.98; CFI = 0.97; RMSEA = 0.049). The χ^2^ difference test between the structural covariances model and the measurement residuals complete invariance model was significant (χ^2^ (9) = 169.996, *p* < 0.05). Overall, these results indicated that the factor loadings, unique variances, and factor variances were different across years, revealing that there were significant differences in psychological well-being among different years in college. Yet it is still hard to determine which of the four years were different, so we conducted a further comparison between different academic years, as reported in Table 5.

The results showed that there was no significant difference in students’ psychological well-being in the freshman and sophomore years, indicated by the measurement weights model, structural covariances model, and measurement residuals complete invariance model (*p* > 0.05). Nevertheless, the mental health states between the sophomore and junior years differed significantly, with *p* < 0.05 for the structural covariances model and the measurement residuals complete invariance model. A similar pattern applied to the comparison between the junior and senior years (*p* < 0.05 for all three models). We also performed difference tests between year 1 and 3, year 1 and 4, and year 2 and 4 for college students as a validation. This indicated that strictly speaking, i.e., by holding factor loadings, variances and residuals as invariant, there were statistically significant differences between the latter three years.

## 4. Discussion

Generally, Chinese college students experienced more mental problems in the freshman and sophomore years. For freshman students, their anxiety may be partly caused by adjustment disorders, especially when students are separated from their parents and friends and have to orient themselves to new environments. Moreover, given the curriculum organization in Chinese universities, which usually set general courses in the first year and introduce more specialized courses from the second year, sophomores may be confronted with intense pressure from study compared to the previous semesters. This may inevitably lead to higher stress and even depression. Our conclusion partly echoed another similar study conducted in the UK, which indicated that once students started university, a great strain was placed on their well-being, with a significant reduction in levels of distress from semester one to semester two being observed in both year one and year three [40]. However, the study in the UK considered university a time of heightened distress, while in China the average DASS scores only indicated certain anxiety problems among college students. We suppose that this disparity can mainly be credited to financial concerns. Previous studies showed that financial pressure was a significant negative linear predictor of psychological health [26,27], and students who dropped out due to financial burden tended to have poor mental and physical health [28]. Therefore, financial difficulties should be deemed as one major source of negative emotions. We take institutional and cultural aspects into account when analyzing the differences in financial pressures on Chinese and British students. From an institutional perspective, Chinese college students usually pay much lower tuition for undergraduate study compared to their counterparts in the UK; meanwhile, the economic aids in China are relatively more accessible regarding both its simpler application process and also its better chance of success. From a cultural perspective, Chinese parents are more commonly seen to pay the costs of higher education of their children, while UK students have to bear the financial burden mostly by themselves.

In addition, the changing trend of students’ psychological well-being in China differs from that in the US. In China, the psychological health of freshmen and sophomores deserves more attention. By contrast, in the US, juniors and seniors scored higher on depression, anxiety, and stress scales compared with freshmen, and students’ top three concerns were academic performance, the pressure to succeed and post-graduation plans [25]. A possible reason behind this difference is that most Chinese universities have strict requirements for students’ national college entrance examination scores, while in the US, college students face intense pressure on meeting graduation requirements, rather than admission thresholds. Therefore, during college, American students need to cope with relatively higher academic stress than Chinese students.

Previous literature on college students’ psychological well-being was generally based on cross-sectional data, so the measurement may not be consistent when comparing students in different years. Our study, using longitudinal data, helps to strengthen the credibility of relevant conclusions. By tracking the same cohort over four years, we confirmed that there were significant differences for students in different years. We attribute the psychological disparity to grade differences, rather than the inapplicability of DASS-21 scale across years, for mainly two reasons. First, the DASS-21 enjoys worldwide recognition in measuring levels of depression, anxiety, and stress, so we assume that it can be applied to examine students in different years [50,51,52,53,54]. Second, we tracked the same group of students for four years during college and measured their mental state with the same psychological well-being scales. Therefore, the differences in the measurement results indicated that students’ mental status experienced changes over time.

## 5. Conclusions

The primary purpose of this study was to examine whether there were differences in the psychological well-being status of Chinese college students as they experienced university life. Through descriptive statistics and multiple group analysis, we can mainly draw the following conclusions.

First, Chinese college students were on average mentally healthy with regards to stress and depression, but they suffered from anxiety beyond normal levels in the first three years.

Second, approximately 20 to 40 percent of college students suffered from different degrees of depression, anxiety, and stress. This was consistent with some relevant previous research, which concluded that roughly one-fifth of students struggled with different levels of mental health problems.

Third, it has been confirmed that there was a disparity in the psychological well-being of students in different years. To be more specific, the highest average scores of depression, anxiety, and stress appeared either in the first or second year. A relatively large ratio of freshmen suffered from anxiety, while, during their sophomore year, students tended to experience more stress and depression. There were significant differences between sophomore, junior and senior students, showing that their mental health situation had improved over time.

Based on the results of this study, we suggest that colleges and universities should provide students with tailored psychological guidance, considering that college students in different years may have differentiated psychological well-being status. Universities may offer proper psychological counseling for freshman students in order to relieve their anxiety and pay special attention to improving the psychological well-being of sophomore students. Furthermore, it is worth noting that the overall optimistic situation of the psychological state of colleges students may be due to their weak perception about changes in their external social environment. In the future, we could study college students’ mental changes after entering the labor market for a certain period. The comparison of their psychological situation at work and during college could be explored to offer more implications on the development of psychological well-being counseling programs in college.

## Figures and Tables

**Figure 1 ijerph-16-02864-f001:**
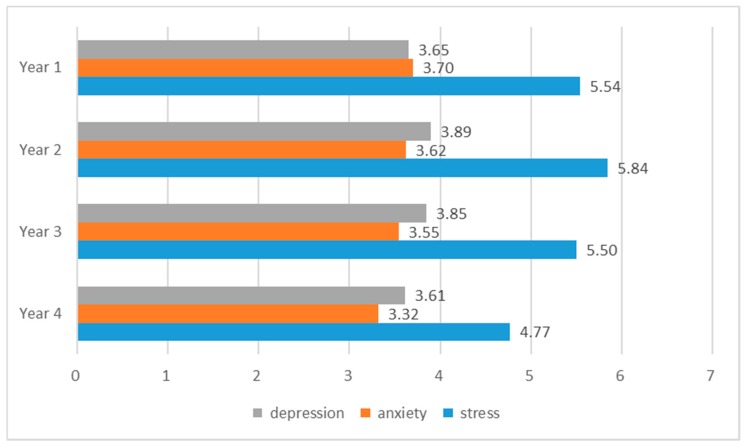
Comparison of means in depression, anxiety and stress across years.

**Figure 2 ijerph-16-02864-f002:**
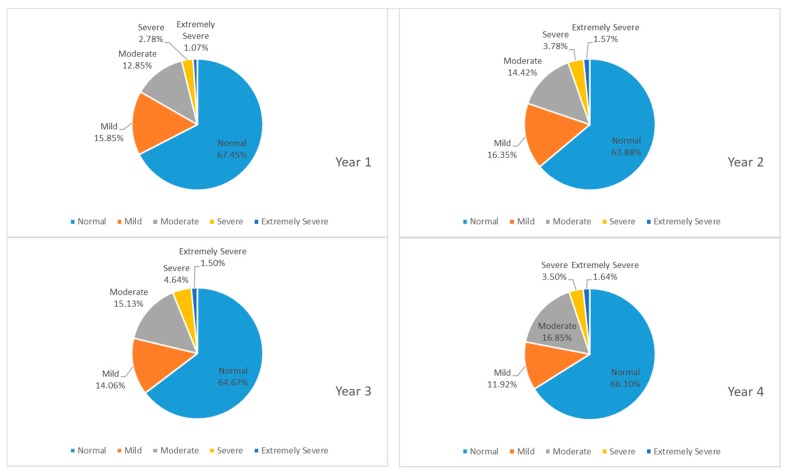
Proportion of participants whose answers on the Depression Anxiety Stress Scales-21 (DASS-21) indicated a normal, mild, moderate, severe or extremely severe amount of depression across years.

**Figure 3 ijerph-16-02864-f003:**
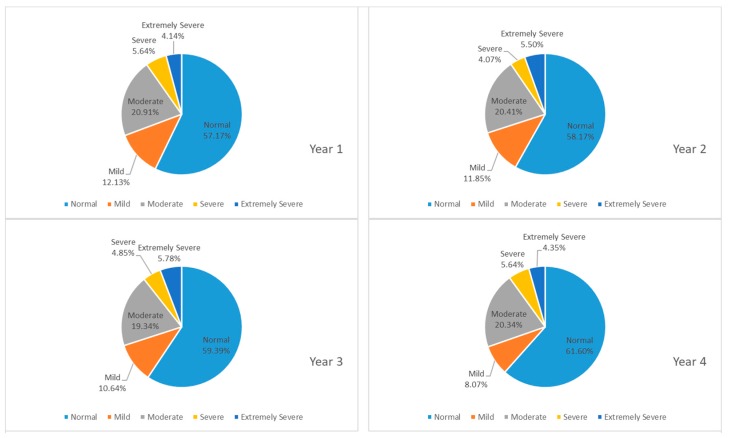
Proportion of participants whose answers on the DASS-21 indicated a normal, mild, moderate, severe or extremely severe amount of anxiety across years.

**Figure 4 ijerph-16-02864-f004:**
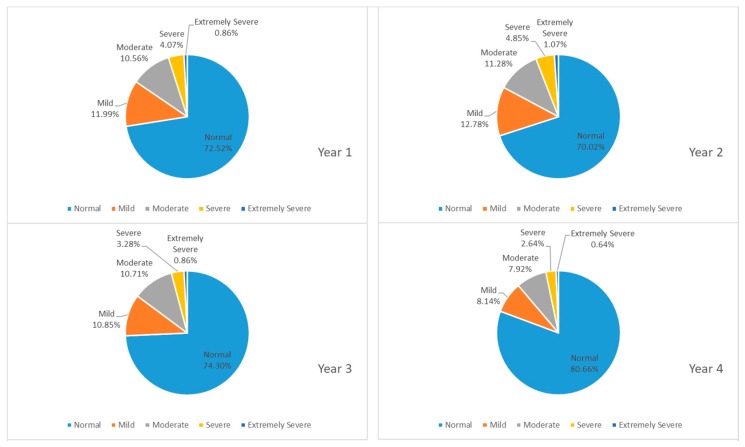
Proportion of participants whose answers on the DASS-21 indicated a normal, mild, moderate, severe or extremely severe amount of stress across years.

**Table 1 ijerph-16-02864-t001:** Sample sizes of different universities.

University	Initial Number of Participants	Effective Sample Size	Proportion (%)
Peking University	246	130	9.28
Renmin University of China	245	147	10.49
Tsinghua University	257	148	10.56
Beihang University	153	89	6.35
Beijing Institute of Technology	157	102	7.28
North China University of Technology	147	91	6.5
Beijing University of Chemical Technology	142	85	6.07
Beijing University of Posts and Telecommunications	128	79	5.64
Beijing Institute of Petroleum and Chemical Technology	136	73	5.21
Beijing University of Agriculture	135	71	5.07
Beijing Language and Culture University	161	82	5.85
Communication University of China	135	69	4.93
Capital University of Economics and Trade	139	74	5.28
Minzu University of China	160	81	5.78
China University of Mining and Technology	132	80	5.71
Total	2473	1401	100

**Table 2 ijerph-16-02864-t002:** Severity labels of depression, anxiety, and stress.

Degree of Severity	Depression	Anxiety	Stress
Normal	0–9	0–7	0–14
Mild	10–13	8–9	15–18
Moderate	14–20	10–14	19–25
Severe	21–27	15–19	26–33
Extremely Severe	28+	20+	34+

**Table 3 ijerph-16-02864-t003:** Descriptive statistics of each item in the Depression Anxiety Stress Scales-21 (DASS-21) across years.

Scale	Item	Year 1		Year 2		Year 3		Year 4	
		Mean	SD	Mean	SD	Mean	SD	Mean	SD
Depression	I couldn’t seem to experience any positive feeling at all	0.45	0.70	0.50	0.73	0.53	0.69	0.53	0.70
	I found it difficult to work up the initiative to do things	0.62	0.78	0.66	0.80	0.61	0.77	0.49	0.68
	I felt that I had nothing to look forward to	0.44	0.73	0.55	0.78	0.52	0.75	0.52	0.71
	I felt down-hearted and blue	0.67	0.75	0.66	0.74	0.65	0.73	0.55	0.67
	I was unable to become enthusiastic about anything	0.44	0.73	0.47	0.73	0.47	0.67	0.48	0.66
	I felt I wasn’t worth much as a person	0.87	0.86	0.88	0.89	0.82	0.82	0.76	0.76
	I felt that life was meaningless	0.16	0.47	0.18	0.49	0.25	0.56	0.28	0.55
Anxiety	I was aware of dryness of my mouth	1.05	0.89	0.98	0.87	0.86	0.76	0.75	0.69
	I experienced breathing difficulty	0.30	0.61	0.34	0.62	0.37	0.62	0.40	0.60
	I experienced trembling	0.22	0.51	0.22	0.53	0.23	0.51	0.27	0.53
	I was worried about situations in which I might panic and make a fool of myself	1.08	0.85	1.01	0.85	0.87	0.82	0.72	0.75
	I felt I was close to panic	0.21	0.53	0.26	0.60	0.32	0.61	0.34	0.61
	I was aware of the action of my heart in the absence of physical exertion	0.36	0.63	0.34	0.64	0.37	0.63	0.38	0.59
	I felt scared without any good reason	0.48	0.72	0.46	0.72	0.52	0.70	0.46	0.66
Stress	I found it hard to wind down	0.73	0.83	0.78	0.85	0.71	0.78	0.59	0.71
	I tended to over-react to situations	0.87	0.85	0.88	0.83	0.86	0.78	0.82	0.70
	I felt that I was using a lot of nervous energy	1.24	0.92	1.30	0.93	1.10	0.82	0.88	0.78
	I found myself getting agitated	0.47	0.66	0.47	0.67	0.53	0.66	0.47	0.63
	I found it difficult to relax	0.79	0.85	0.88	0.89	0.84	0.82	0.72	0.74
	I was intolerant of anything that kept me from getting on with what I was doing	0.67	0.73	0.67	0.77	0.66	0.72	0.56	0.68
	I felt that I was rather touchy	0.77	0.83	0.86	0.86	0.80	0.78	0.72	0.76
Severity Degree Scores	Depression	7.30		7.79		7.70		7.22	
	Anxiety	7.40		7.24		7.10		6.63	
	Stress	11.07		11.68		11.00		9.53	

Note: SD = Standard Deviation.

**Table 4 ijerph-16-02864-t004:** Standardized parameter estimates of models.

Model	Unconstrained				Measurement Weights				Structural Covariances				Measurement Residuals
Year	1	2	3	4	1	2	3	4	1	2	3	4	1/2/3/4
λ1	0.73	0.72	0.80	0.87	0.75	0.75	0.80	0.86	0.78	0.76	0.79	0.83	0.79
λ2	0.78	0.83	0.83	0.90	0.81	0.83	0.84	0.89	0.84	0.84	0.83	0.87	0.84
λ3	0.82	0.81	0.87	0.87	0.77	0.79	0.86	0.89	0.81	0.81	0.86	0.88	0.84
e1	0.34	0.34	0.24	0.24	0.41	0.37	0.26	0.21	0.35	0.35	0.26	0.23	0.29
e2	0.39	0.32	0.31	0.19	0.34	0.31	0.29	0.21	0.29	0.30	0.31	0.25	0.29
e3	0.46	0.48	0.37	0.25	0.44	0.44	0.37	0.27	0.39	0.42	0.38	0.31	0.38

Note: λ1 = stress factor loadings; λ2 = anxiety factor loadings; λ3 = depression factor loadings; e = error variance.

**Table 5 ijerph-16-02864-t005:** Comparison of model results over the four academic years.

Comparison	Model	Degree of Freedom	χ^2^	*p*
Year 1 vs. Year 2	Measurement weights	2	4.083	0.130
	Structural covariances	1	3.820	0.051
	Measurement residuals	3	3.911	0.271
Year 2 vs. Year 3	Measurement weights	2	3.657	0.161
	Structural covariances	1	4.939	0.026
	Measurement residuals	3	31.315	0.000
Year 3 vs. Year 4	Measurement weights	2	15.006	0.001
	Structural covariances	1	4.323	0.038
	Measurement residuals	3	53.123	0.000
Year 1 vs. Year 3	Measurement weights	2	5.238	0.073
	Structural covariances	1	17.366	0.000
	Measurement residuals	3	25.110	0.000
Year 1 vs. Year 4	Measurement weights	2	30.540	0.000
	Structural covariances	1	44.160	0.000
	Measurement residuals	3	111.691	0.000
Year 2 vs. Year 4	Measurement weights	2	3.657	0.161
	Structural covariances	1	4.939	0.026
	Measurement residuals	3	31.315	0.000

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
