# Peer review of "Changes in Undergraduate Students’ Psychological Well-Being as They Experience University Life"

_ijerph, 2019, doi:10.3390/ijerph16162864_

Round 1

Reviewer 1 Report

Thank you for inviting me to review the paper “Changes in undergraduate students’ psychological well-being as they experience university life”. This is an important paper with sound methodology. I recommend publication in IJERPH. I have the following comments for minor revision.

1.     Line 43-44, the authors mentioned that “Ohrnberger et al. (2017) studied 10,693 individuals over the age of 50, using lifestyle choices and social capital in a mediation framework to analyze the direct and indirect impacts of the past mental and physical health status.” I think this reference is not too relevant as this paper is about university students in China. I suggest to refer to the following landmark papers in Asia to illustrate mental and physical health status in young people. Please replace the above statement with the following statements:

Previous studies found that adverse physical health status (e.g. obesity, asthma, skin diseases, hepatitis, heart diseases) has an impact on mental health in young people and adults (Quek et al 2017, Lu et al 2014, Nguyen et al 2019, Vu et al 2019, Ho et al 2018).

References:

Quek YH et al Exploring the association between childhood and adolescent obesity and depression: a meta-analysis. Obes Rev. 2017 Jul;18(7):742-75

Lu Y et al Psychiatric comorbidities in Asian adolescent asthma patients and the contributions of neuroticism and perceived stress. J Adolesc Health. 2014 Aug;55(2):267-75.

Nguyen SH et al Health-Related Quality of Life Impairment among Patients with Different Skin Diseases in Vietnam: A Cross-Sectional Study. Int J Environ Res Public Health. 2019 Jan 23;16(3).

Vu TTM et al Socioeconomic Vulnerability to Depressive Symptoms in Patients with Chronic Hepatitis B. Int J Environ Res Public Health. 2019 Jan 17;16(2). pii: E255. 

Ho RCM et al Factors Associated with the Risk of Developing Coronary Artery Disease in Medicated Patients with Major Depressive Disorder. Int J Environ Res Public Health. 2018 Sep 21;15(10). 

2.     Line 50, the authors stated that “Emotional well-being, as part of mental health, incorporates happiness, interests in life, satisfaction [4, 5], and the ability to recognize, express and regulate one’s emotions. I suggest to authors to add quality of life, especially for Chinese. Please amend the statement as follows:

as part of mental health, incorporates happiness, interests in life, satisfaction and quality of life [4, 5, Tan et al 2015],

Reference:

Tan SH et al Determining the quality of life of depressed patients in Singapore through a multiple mediation framework. Asian J Psychiatr. 2015 Dec;18:22-30.

3.     Line 54, the authors stated that “The widely recognized DASS-21 scale is a measure of negative emotions in depression, anxiety, and stress.” This statement lacks good references in Chinese population. Please add the following references from IJERPH to support such claim.

The widely recognized DASS-21 scale is a measure of negative emotions in depression, anxiety, and stress in Chinese (Quek et al 2018, Ho CS et al 2019).

References:

Quek TC et al Misophonia in Singaporean Psychiatric Patients: A Cross-Sectional Study. Int J Environ Res Public Health. 2018 Jul 4;15(7). pii: E1410. 

Ho CSH et al Relationship of Anxiety and Depression with Respiratory Symptoms: Comparison between Depressed and Non-Depressed Smokers in Singapore. Int J Environ Res Public Health. 2019 Jan 8;16(1).

4.     Line 61, the authors stated “A study by the Anxiety and Depression Association of America found that 70 percent of American adults suffered from at least moderate anxiety or stress every day [10]” It is important to comment on global prevalence of depression. I suggest the authors to quote the prevalence figure of a recent study on “the Prevalence of Depression in the Community from 30 Countries between 1994 and 2014. (Journal: Scientific Report)” Please find this landmark study on Pubmed.

5.     Line 68, the authors stated “Likewise, a study from Egypt indicated 68 that among 442 medical students of Fayoum University with an average age of 22.15±1.9 years, 60.8  percent, 64.3 percent, and 62.4 percent of the sample suffered from depression, anxiety, and stress, respectively [17].” This reference is not representative as it is based on Egypt only. The authors should also cite a reference representing all medical students from Asia. Please quote the prevalence figure from this study “Mental health issues amongst medical students in Asia: a systematic review [2000-2015].” Please find this landmark study on Pubmed.

6.     Line 84, the authors stated that “College is a crucial period of life for students to shape proper values, worldviews, and outlooks of life”. I suggest to add resilience as it is relevant to Chinese college students. Please modify the statement as follows:

            College is a crucial period of life for students to shape proper values, worldviews, outlooks of life and resilience (Ramsay et al 2015).

            Reference

            Ramsay JE et al Divergent pathways to influence: Cognition and behavior differentially                 mediate the effects of optimism on physical and mental quality of life in Chinese university             students. J Health Psychol. 2015 Jul;20(7):963-73.

7.     For Table 1, can the authors provide information about the disciplines of their study? If such information is not available, please state in the limitations.

8.     For Table 3, the authors label each item as Stress 01, 02, Anxiety 01, 02. These are not meaningful for the readers. Can the authors put in actual descriptive term for each item?

9.     Line 325, the authors mentioned that “The corresponding proportion of students with psychological problems appeared to be lower in the United States [15], while the situation in Egyptian colleges was much more severe concerning the depression, anxiety and stress scales [17]”. United States and Egypt do not represent the whole word. The authors should compare with the prevalence in landmark meta-analyses, “Prevalence of depression among nursing students: A systematic review and meta-analysis (Journal: Nurse Education Today)” and “Prevalence of depression amongst medical students: a meta-analysis (Journal: Medical Education)”.

Author Response

Response to Reviewer 1 Comments

Dear Reviewer,

Thanks so much for your careful reading of our manuscript and we really appreciate your suggestions that allowed us to greatly improve the quality of this paper. In the following, we have addressed your comments point by point. Throughout, the reviewer’ comments are in black font, and our responses in red. Should there be any further concerns or questions, please do not hesitate to contact us.

With Best Regards,

Xinqiao Liu, Siqing Ping, Wenjuan Gao*

General Comments:

Thank you for inviting me to review the paper “Changes in undergraduate students’ psychological well-being as they experience university life”. This is an important paper with sound methodology. I recommend publication in IJERPH. I have the following comments for minor revision. 

Point 1: Line 43-44, the authors mentioned that “Ohrnberger et al. (2017) studied 10,693 individuals over the age of 50, using lifestyle choices and social capital in a mediation framework to analyze the direct and indirect impacts of the past mental and physical health status.” I think this reference is not too relevant as this paper is about university students in China. I suggest to refer to the following landmark papers in Asia to illustrate mental and physical health status in young people.

Response 1: Thank you for your constructive comments. We have deleted the irrelevant reference, and included the landmark papers as suggested. The revised content is shown below.

Line 41-43: Previous studies also found that adverse physical health status (e.g. obesity, asthma, skin diseases, hepatitis, heart diseases) has an impact on mental health in young people and adults [1-5].

References:

[1] Quek, Y.H.; Tam, W.W.; Zhang, M.W.; Ho, R.C. Exploring the association between childhood and adolescent obesity and depression: a metaanalysis. Obesity reviews 2017, 18, 742-754

[2] Lu, Y.; Ho, R.; Lim, T.K.; Kuan, W.S.; Goh, D.Y.T.; Mahadevan, M.; Sim, T.B.; Ng, T.-P.; van Bever, H.P. Psychiatric comorbidities in Asian adolescent asthma patients and the contributions of neuroticism and perceived stress. Journal of Adolescent Health 2014, 55, 267-275.

[3] Nguyen, S.H.; Nguyen, L.H.; Vu, G.T.; Nguyen, C.T.; Le, T.H.T.; Tran, B.X.; Latkin, C.A.; Ho, C.S.; Ho, R. Health-Related Quality of Life Impairment among Patients with Different Skin Diseases in Vietnam: A Cross-Sectional Study. International journal of environmental research and public health 2019, 16, 305.

[4] Vu, T.T.M.; Le, T.V.; Dang, A.K.; Nguyen, L.H.; Nguyen, B.C.; Tran, B.X.; Latkin, C.A.; Ho, C.S.; Ho, R. Socioeconomic vulnerability to depressive symptoms in patients with chronic hepatitis B. International journal of environmental research and public health 2019, 16, 255.

[5] Ho, R.; Chua, A.; Tran, B.; Choo, C.; Husain, S.; Vu, G.; McIntyre, R.; Ho, C. Factors associated with the risk of developing coronary artery disease in medicated patients with major depressive disorder. International journal of environmental research and public health 2018, 15, 2073.

Point 2: Line 50, the authors stated that “Emotional well-being, as part of mental health, incorporates happiness, interests in life, satisfaction [4, 5], and the ability to recognize, express and regulate one’s emotions. I suggest to authors to add quality of life, especially for Chinese.

Response 2: we have added quality of life as suggested. The revised content is as follows:

Line 43-44:  …Emotional well-being, as part of mental health, incorporates happiness, interests in life, satisfaction and quality of life [6-8]…

Reference:

[8] Tan, S.H.; Tang, C.; Ng, W.W.; Ho, C.S.; Ho, R.C. Determining the quality of life of depressed patients in Singapore through a multiple mediation framework. Asian journal of psychiatry 2015, 18, 22-30.

Point 3: Line 54, the authors stated that “The widely recognized DASS-21 scale is a measure of negative emotions in depression, anxiety, and stress.” This statement lacks good references in Chinese population. Please add the following references from IJERPH to support such claim.

Response 3: We have added related references from IJERPH to support this claim.

Line 46-47: The widely recognized DASS-21 scale is a measure of negative emotions in depression, anxiety, and stress in Chinese [10,11].  

References:

[10] Quek, T.; Ho, C.; Choo, C.; Nguyen, L.; Tran, B.; Ho, R. Misophonia in Singaporean psychiatric patients: a cross-sectional study. International journal of environmental research and public health 2018, 15, 1410.

[11] Ho, C.S.; Tan, E.L.; Ho, R.; Chiu, M.Y. Relationship of anxiety and depression with respiratory symptoms: comparison between depressed and non-depressed smokers in Singapore. International journal of environmental research and public health 2019, 16, 163.

Point 4: Line 61, the authors stated “A study by the Anxiety and Depression Association of America found that 70 percent of American adults suffered from at least moderate anxiety or stress every day [10]” It is important to comment on global prevalence of depression. I suggest the authors to quote the prevalence figure of a recent study on “the Prevalence of Depression in the Community from 30 Countries between 1994 and 2014. (Journal: Scientific Report)” Please find this landmark study on Pubmed.

Response 4: We have replaced the previous reference with the recent study you mentioned.

Line 53-57: By combining the data of over 1 million participants from 30 countries between 1994 and 2014, relevant research found that the global aggregate point prevalence of depression was 12.9 percent. South America had the highest depression rate, reaching up to 20.6 percent, followed by Asia (16.7%), North America (13.4%), Europe (11.9%), Africa (11.5%), Australia (7.3%) [14].

Reference:

[14] Lim, G.Y.; Tam, W.W.; Lu, Y.; Ho, C.S.; Zhang, M.W.; Ho, R.C. Prevalence of Depression in the Community from 30 Countries between 1994 and 2014. Scientific reports 2018, 8, 2861.

Point 5: Line 68, the authors stated “Likewise, a study from Egypt indicated 68 that among 442 medical students of Fayoum University with an average age of 22.15±1.9 years, 60.8  percent, 64.3 percent, and 62.4 percent of the sample suffered from depression, anxiety, and stress, respectively [17].” This reference is not representative as it is based on Egypt only. The authors should also cite a reference representing all medical students from Asia. Please quote the prevalence figure from this study “Mental health issues amongst medical students in Asia: a systematic review [2000-2015].” Please find this landmark study on Pubmed.

Response 5: Considering the article length and reference representativeness, we have deleted the reference from Egypt, and added the reference you recommended.

Line 60-62: Another related study showed that in Asia, 11.0 percent of college students suffered from depression, while the aggregate prevalence of anxiety disorders was 7.04 percent [19].

Reference:

[19] Cuttilan, A.N.; Sayampanathan, A.A.; Ho, R.C.-M. Mental health issues amongst medical students in Asia: a systematic review [2000–2015]. Annals of translational medicine 2016, 4.

Point 6: Line 84, the authors stated that “College is a crucial period of life for students to shape proper values, worldviews, and outlooks of life”. I suggest to add resilience as it is relevant to Chinese college students.

Response 6: Thanks for your suggestion. We have added resilience to the statement.

Line 76-77: College is a crucial period of life for students to shape proper values, worldviews, outlooks of life, and resilience [30].

Reference:

[30] Ramsay, J.E.; Yang, F.; Pang, J.S.; Lai, C.-M.; Ho, R.C.; Mak, K.-K. Divergent pathways to influence: Cognition and behavior differentially mediate the effects of optimism on physical and mental quality of life in Chinese university students. Journal of health psychology 2015, 20, 963-973.

Point 7: For Table 1, can the authors provide information about the disciplines of their study? If such information is not available, please state in the limitations.

Response 7: We have included relevant description about disciplines of their study (see Line 127-133). In fact, students of our sample come from a variety of academic disciplines and we have mainly divided their majors into four categories, i.e., social sciences, humanities, science and engineering and the others (including students who had no specific major upon entering college and a small number of unanswered participants).

Line 127-133: As for the academic disciplines of students, we have mainly divided their majors into four categories, namely social sciences, humanities, science and engineering and the others (including students who had no specific major upon entering college and a small number of unanswered participants). There were 426 students and 193 students majoring in social science and humanities, respectively, accounting for 30.41 percent and 13.78 percent of the sample. Moreover, 757 students studied in the discipline of science and engineering, representing 54.03 percent of the total; and another 25 students were classified as the others (1.78%).

Point 8: For Table 3, the authors label each item as Stress 01, 02, Anxiety 01, 02. These are not meaningful for the readers. Can the authors put in actual descriptive term for each item?

Response 8: We have replaced the labels as Stress 01, 02, Anxiety 01, 02 with the description of corresponding items, as indicated in Table 3 in the revised manuscript.

Point 9: Line 325, the authors mentioned that “The corresponding proportion of students with psychological problems appeared to be lower in the United States [15], while the situation in Egyptian colleges was much more severe concerning the depression, anxiety and stress scales [17]”. United States and Egypt do not represent the whole word. The authors should compare with the prevalence in landmark meta-analyses, “Prevalence of depression among nursing students: A systematic review and meta-analysis (Journal: Nurse Education Today)” and “Prevalence of depression amongst medical students: a meta-analysis (Journal: Medical Education)”.

Response 9: Thank you for the advice. We have removed the reference from United States, Egypt, and Ecuador, and added the global prevalence as you recommended. On second thoughts, we included the statements in the introduction part, and the revised content are shown below:

 Line 58-60: A meta-analysis study indicated that the worldwide prevalence of depression among medical students was 34.0 percent, with the highest proportion reported in Asia at 43.0 percent [18].

Line 88-90: Globally, Puthran, et al. (2016) found that medical freshman students had the highest rates of depression at 33.5 percent, which then experienced a significant decline over time to 20.5 percent before graduation [38].

Reference:

[18] Tung, Y.-J.; Lo, K.K.; Ho, R.C.; Tam, W.S.W. Prevalence of depression among nursing students: a systematic review and meta-analysis. Nurse education today 2018, 63, 119-129.

[38] Puthran, R.; Zhang, M.W.; Tam, W.W.; Ho, R.C. Prevalence of depression amongst medical students: a metaanalysis. Medical education 2016, 50, 456-468.

Reviewer 2 Report

Overall, I think that the article is very interesting, as well as the dataset. A longitudinal study on such a large sample is not common to found. Also for this reason, I think that the analysis, graphic presentation and discussion could be made more clear, thus becoming more powerful. Also, I think that in some point the sentences could be more fluid, as some time the meaning is not totally clear.

I have put notes in the pdf file of the article. I attach here the PDF with my observation.

Author Response

Dear Reviewer,

Thanks so much for your careful reading of our manuscript, and we really appreciate your constructive comments and suggestions that allowed us to greatly improve the quality of this paper. We have addressed your comments point by point as follows. Please see the attachment. Throughout, the reviewer’ comments are in black font, and our responses in red. Should there be any further concerns or questions, please do not hesitate to contact us.

With Best Regards,

Xinqiao Liu, Siqing Ping, Wenjuan Gao*

Reviewer 3 Report

This article highlights the important topic of college students’ mental health in China and makes a case for when prevention and intervention efforts may be needed most. However, there are a number of issues that require attention. These include:

The entire manuscript requires copyediting. There are numerous grammatical/mechanical issues throughout that impacts the overall quality and clarity of the article.

The introduction could be organized better. The content does not read smoothly and seems to include discussion of findings from the literature that seems unnecessary and distracts from the key message. Some content may even be somewhat unsuitable given the focus on undergraduate students (e.g., Wahed & Hassan, 2017).

Sources could be stronger. One reference was a dissertation (Wolfram, 2010), which was used as the citation for the statement, “In the US, nearly 10 percent of college students were diagnosed with depression.” Consider other (primary) sources for prevalence data.

There is little detail on the BCSPS provided. No information was provided about when the four waves of data collection were conducted or how many students the survey initially targeted.

The authors used a quote from the WHO that states that mental health is not just the absence of disease/infirmity. The authors also discussed other factors (e.g., happiness, interests, life satisfaction; things that they did not measure) as part of being mentally well. Despite these points, the authors refer to students with normal scores on the stress and depression scales as “mentally healthy,” which seems to somewhat contradict their own discussion about what mental well-being is.

The descriptive statistics for the DASS 21 results did not seem to provide strong support for the authors’ firm conclusion that first and second year college students require greater attention to address mental health challenges compared to third and fourth year students.

It is not clear to me why the authors decided to use multigroup CFA for this study. They described an analysis to look at whether the DASS 21 means the same thing to students in the different grade level groups. I thought they were simply looking to see if significant differences existed between the grade level groups? Seems more like an analysis of variance study.

Author Response

Response to Reviewer 2 Comments

Dear Reviewer,

We highly appreciate your thoughtful and detailed suggestions. We have addressed the comments point by point and responded to them accordingly. Throughout, the reviewer’ comments are in black font, and our responses in red. We think the paper has been greatly improved after revision. Please check the attached PDF documents if there are problems with the format display. Please reach out to us if there are any questions or concerns.

With Best Regards,

Xinqiao Liu, Siqing Ping, Wenjuan Gao*

Comments and Suggestions for Authors

This article highlights the important topic of college students’ mental health in China and makes a case for when prevention and intervention efforts may be needed most. However, there are a number of issues that require attention. These include:

Point 1: The entire manuscript requires copyediting. There are numerous grammatical/ mechanical issues throughout that impacts the overall quality and clarity of the article.

Response 1:  With regard to the language editing issues, we have had our manuscript checked by some native English-speaking colleagues and have made improvements on grammar, spelling, and punctuation, and also have adjusted the logics and style. If further editing is still needed, please kindly notify us and we may address it by seeking professional editing service of MDPI.

Point 2: The introduction could be organized better. The content does not read smoothly and seems to include discussion of findings from the literature that seems unnecessary and distracts from the key message. Some content may even be somewhat unsuitable given the focus on undergraduate students (e.g., Wahed & Hassan, 2017).

Response 2: Thank you for your constructive suggestion. We have reorganised the introduction, removing unnecessary information and adding some reprehensive and supportive references. For example, we deleted the following references:

Line 43-44: “Ohrnberger et al. (2017) studied 10,693 individuals over the age of 50, using lifestyle choices and social capital in a mediation framework to analyse the direct and indirect impacts of the past mental and physical health status.”

Line 52-53:  Globally, the number of individuals suffering from anxiety and/or depression rose by 50% from 1990 to 2013; totalling 615 million individuals in 2013[7].

Line 67-72:  In the US, nearly 10 percent of college students were diagnosed with depression [15], and almost half of the students met the criteria of DSM axis I disorder [16]. Likewise, a study from Egypt indicated that among 442 medical students of Fayoum University with an average age of 22.15±1.9 years, 60.8 percent, 64.3 percent, and 62.4 percent of the sample suffered from depression, anxiety, and stress, respectively [17]. Wattick et al.2018found that students’ mean number of depressed days over the past 30 days was 9.67 ± 8.80, and of anxious days, 14.1 ± 10.03[18].

Line 122-131: As for the discrepancy in mental status of college students and their non-college-attending peers, the study by Blanco et al. (2008) in the United States showed that nearly half of college-aged individuals had mental disorder, with no significant difference between college and non-college-attending individuals; though the risk of alcohol use disorder for college students turned out to be much higher than their non-college counterparts [16]. Nevertheless, another related study by Stewart-Brown et al. (2000) drew a quite different conclusion after comparing students from three UK institutes of higher education and their equivalent 18-to-34 age group in the local population. The study employed the SF-36 health status measurement and indicated that the health of college students, especially their emotional health, was more serious relative to their peers, and the emotional pressure mainly came from study or work problems and money [34].

Furthermore, we added some landmark meta-analysis studies:

Line 41-43: Previous studies also found that adverse physical health status (e.g. obesity, asthma, skin diseases, hepatitis, heart diseases) has an impact on mental health in young people and adults [1-5].

Line 53-57: By combining the data of over 1 million participants from 30 countries between 1994 and 2014, relevant research found that the global aggregate point prevalence of depression was 12.9 percent. South America had the highest depression rate, reaching up to 20.6 percent, followed by Asia (16.7%), North America (13.4%), Europe (11.9%), Africa (11.5%), Australia (7.3%) [14].

Line 58-60: A meta-analysis study indicated that the worldwide prevalence of depression among medical students was 34.0 percent, with the highest proportion reported in Asia at 43.0 percent [18].

Line 60-62: Another related study showed that in Asia, 11.0 percent of college students suffered from depression, while the aggregate prevalence of anxiety disorders was 7.04 percent [19].

Line 88-90: Globally, Puthran, et al. (2016) found that medical freshman students had the highest rates of depression at 33.5 percent, which then experienced a significant decline over time to 20.5 percent before graduation [38].

Point 3: Sources could be stronger. One reference was a dissertation (Wolfram, 2010), which was used as the citation for the statement, “In the US, nearly 10 percent of college students were diagnosed with depression.” Consider other (primary) sources for prevalence data.

Response 3: We have replaced this citation with some more representative studies to make our sources stronger. Here instead of using the original reference, we chose to compare the prevalence in landmark meta-analyses which are shown as follows. In addition, all other references have been under scrutiny to ensure the reliability and validity.

Line 58-62: A meta-analysis study indicated that the worldwide prevalence of depression among medical students was 34.0 percent, with the highest proportion reported in Asia at 43.0 percent [18]. Another related study showed that in Asia, 11.0 percent of college students suffered from depression, while the aggregate prevalence of anxiety disorders was 7.04 percent [19].

Point 4: There is little detail on the BCSPS provided. No information was provided about when the four waves of data collection were conducted or how many students the survey initially targeted.

Response 4: In the participants and procedure section, we have added detailed information concerning the time of data collection and the follow-up rates (see the following extract). Moreover, in Table 1, we have included the initial numbers of participants from different universities in the first round of survey as well as the final effective sample size.

Line 115-117: Participants of the survey were students admitted to 15 universities in the years of 2006 and 2008; and these two cohorts have been tracked for four consecutive years from June 2009 to June 2012.

Line 121-133: In order to explore changes of students’ mental states across years during college, the 2008 cohort was selected and analysed in our study. The initial numbers of participants from different universities in the first round has been listed in Table 1, and the follow-up rate of the latter three rounds was 95.27%, 94.66%, and 90.58%, respectively, with very few sample losses [38]. After removing invalid questionnaires, the effective sample size of this panel study was 1401, including 650 females and 751 males. The proportion of samples from different universities are also shown in Table 1. As for the academic disciplines of students, we have mainly divided their majors into four categories, namely social sciences, humanities, science and engineering and the others (including students who had no specific major upon entering college and a small number of unanswered participants). There were 426 students and 193 students majoring in social science and humanities, respectively, accounting for 30.41 percent and 13.78 percent of the sample. Moreover, 757 students studied in the discipline of science and engineering, representing 54.03 percent of the total; and another 25 students were classified as the others (1.78%).

Table 1. The sample size of different universities.

University

Initial Number of Participants

Effective Sample Size

Proportion (%)

Peking University

246

130

9.28

Renmin University of China

245

147

10.49

Tsinghua University

257

148

10.56

Beihang University

153

89

6.35

Beijing Institute of Technology

157

102

7.28

North China University of Technology

147

91

6.5

Beijing University of Chemical Technology

142

85

6.07

Beijing University of Posts and Telecommunications

128

79

5.64

Beijing Institute of Petroleum and Chemical Technology

136

73

5.21

Beijing University of Agriculture

135

71

5.07

Beijing Language and Culture University

161

82

5.85

Communication University of China

135

69

4.93

Capital University of Economics and Trade

139

74

5.28

Minzu University of China

160

81

5.78

China University of Mining and Technology

132

80

5.71

Total

2473

1401

100

Point 5: The authors used a quote from the WHO that states that mental health is not just the absence of disease/infirmity. The authors also discussed other factors (e.g., happiness, interests, life satisfaction; things that they did not measure) as part of being mentally well. Despite these points, the authors refer to students with normal scores on the stress and depression scales as “mentally healthy,” which seems to somewhat contradict their own discussion about what mental well-being is.

Response 5: We feel really grateful for the reviewer’s careful reading of our manuscript. Indeed, mental well-being refers to many aspects as stated in the definition by WHO. From previous literature, it can be noticed that depression, anxiety, and stress are the most important indicators of mental health measures. In order to avoid such misunderstanding, we have removed the WHO quote from our manuscript and adjusted the logics in order to focus more on our research topics.

Point 6: The descriptive statistics for the DASS 21 results did not seem to provide strong support for the authors’ firm conclusion that first and second year college students require greater attention to address mental health challenges compared to third and fourth year students.

Response 6: Figure 1 compares the average scores of the depression, anxiety and stress scales. Specifically, it can be found that the mean scores of stress scale during the four academic years were 5.54, 5.84, 5.50 and 4.77, respectively. Similarly, the average scores of anxiety scale were 3.70, 3.62, 3.55, 3.32 over time. These numbers indicate that students scored higher in freshman and sophomore years than the latter two years, though the differences appeared to be not so evident. In this way, we compared the proportions of students with different degrees of depression, anxiety and stress. In Figure 2, we can tell that 72.52 percent of freshmen experienced normal levels of stress, and by the last year of college, this ratio went up to 80.66 percent. Furthermore, student who fell in the normal range of anxiety accounted for 57.17 percent in the freshmen year, and the proportion increased to 61.6 percent. Therefore, both the average levels of college students’ stress, anxiety and depression, and the proportions of students with normal levels of the three scales support our conclusion. In addition, we have employed the multiple group analysis to identify whether there existed significant differences across years in college to further corroborate the analysis of descriptive statistics.

Point 7: It is not clear to me why the authors decided to use multigroup CFA for this study. They described an analysis to look at whether the DASS 21 means the same thing to students in the different grade level groups. I thought they were simply looking to see if significant differences existed between the grade level groups? Seems more like an analysis of variance study.

Response 7: To our knowledge, an analysis of variance study is used to compare differences in group means of a sample, while multigroup CFA allows not only a comparison of path coefficients between groups, but also a comparison of means as well as intercepts. Indeed, there are mainly three reasons that we chose to adopt multigroup CFA rather than an analysis of variance. First, an analysis of variance only identifies whether the means across grades are equal, while multigroup CFA can further compare their factor loadings, covariances and residuals, which may improve the accuracy of our analysis. Second, Figure 1 indicates that the mean scores of stress, anxiety, and depression were indeed greater in freshman and sophomore years than in the last two years, but the differences were not so evident; thus, it would not be convincing to conclude there existed significant differences across years only by comparing the average scores. Third, the goal of this section is not just to analyse whether the scales of stress, anxiety, and depression differed separately across grades; instead, we aim to explore differences in overall psychological well-being for students of different years. The approach of variance analysis essentially compares sums of the means of stress, anxiety and depression, which lacks theoretical rigor compared with the method of multigroup CFA. As shown in Table R1, the results of variance analysis are not consistent with results of multiple group CFA, and we think it would be more appropriate to employ multigroup CFA in this situation.

Table R1. Results of variance analysis

ModelANOVA

Comparison

P value

Significance

Stress

Year 1 vs. Year 2

0.020

*

Year 2 vs. Year 3

0.015

*

Year 3 vs. Year 4

0.000

*

Year 1 vs. Year 3

0.454

N

Year 1 vs. Year 4

0.000

*

Year 2 vs. Year 4

0.000

*

Anxiety

Year 1 vs. Year 2

0.050

*

Year 2 vs. Year 3

0.075

N

Year 3 vs. Year 4

0.001

*

Year 1 vs. Year 3

0.001

*

Year 1 vs. Year 4

0.000

*

Year 2 vs. Year 4

0.000

*

Depression

Year 1 vs. Year 2

0.059

N

Year 2 vs. Year 3

0.147

N

Year 3 vs. Year 4

0.008

*

Year 1 vs. Year 3

0.303

N

Year 1 vs. Year 4

0.030

*

Year 2 vs. Year 4

0.000

*

Mental Health

Year 1 vs. Year 2

0.162

N

Year 2 vs. Year 3

0.026

*

Year 3 vs. Year 4

0.000

*

Year 1 vs. Year 3

0.169

N

Year 1 vs. Year 4

0.000

*

Year 2 vs. Year 4

0.000

*

Note* P<0.05

Round 2

Reviewer 3 Report

Overall, the authors have addressed several of the issues in my previous review but a couple remain. With some revision to the following items, I would recommend publication.

The grammar/mechanics and organization have improved. However, there are errors still present and organization of the literature review can be further improved.

The authors indicate that while their results show that students scored as normal on depression and stress, the average anxiety scores of 7.4, 7.24, and 7.1 indicated that students suffered from mild stress. However, the DASS-21 scoring guide from the developers indicate that an anxiety of score would need to meet a threshold score of 8 before it can be classified as “mild.” This conclusion/argument is rather weak and should be further addressed. Even if the authors were to round the average anxiety scores to the nearest whole number, the scores would not meet the minimum score for classification as “mild.”

Author Response

Dear Reviewer,

Thanks so much for your careful reading of our manuscript, and we really appreciate your constructive suggestions that allowed us to improve the quality of this paper. We have addressed your comments point by point as follows. Throughout, the reviewer’ comments are in black font, and our responses in red. Should there be any further concerns or questions, please do not hesitate to contact us.

With Best Regards,

Xinqiao Liu, Siqing Ping, Wenjuan Gao*

General Comments:

Overall, the authors have addressed several of the issues in my previous review but a couple remain. With some revision to the following items, I would recommend publication.

Point 1: The grammar/mechanics and organization have improved. However, there are errors still present and organization of the literature review can be further improved.

Response 1: Thanks so much for your suggestion. We have carefully checked errors in the manuscript and made corresponding revisions.

As for the organization of the literature review, we have made some adjustments to create a more logical and smooth flow. In general, the first paragraph introduces the significance of mental health, and Depression Anxiety Stress Scale as an effective measure of psychological distress; the second paragraph presents some statistics of mental problems among the general population, and more importantly, among college students in different contexts; the third paragraph further emphasizes the importance of college period and explores the influencing factors and mechanism of students’ psychological well-being;  the fourth paragraph points out the research gap that existing relevant studies used cross-sectional data, and briefly states the few studies using longitudinal design; while the last paragraph explains the significance of our research.

Point 2: The authors indicate that while their results show that students scored as normal on depression and stress, the average anxiety scores of 7.4, 7.24, and 7.1 indicated that students suffered from mild stress. However, the DASS-21 scoring guide from the developers indicate that an anxiety of score would need to meet a threshold score of 8 before it can be classified as “mild.” This conclusion/argument is rather weak and should be further addressed. Even if the authors were to round the average anxiety scores to the nearest whole number, the scores would not meet the minimum score for classification as “mild.”

Response 2: We agree to the reviewer that there might be some inaccuracy in our argument. From our perspective, the DASS-21 scoring guide is applicable at the individual level, so all the cutting off values are shown as integers. However, the average anxiety scores of the first three years turned out to be all decimals (7.4, 7.24, and 7.1). In this case, we believed that if the mean scores overtook the normal threshold 7, then the average anxiety levels should be considered as beyond normal range, and that’s why we classified them as “mild”. After further discussion, we have realized our expressions were indeed not accurate, as you mentioned. Therefore, we have modified our statements in order to reflect the results more precisely and properly.

Line 23-24: Chinese college students in general were mentally healthy with regard to depression and stress, but their average anxiety levels were beyond normal in the first three years.

Line 165-166:  …on average Chinese college students suffered from above-normal levels of anxiety in the first three years…

Line 324-325: First, Chinese college students were on average mentally healthy regarding stress and depression, but they suffered from anxiety beyond normal levels in the first three years.